# Ball Bearing Fault Diagnosis Using Recurrence Analysis

**DOI:** 10.3390/ma15175940

**Published:** 2022-08-27

**Authors:** Krzysztof Kecik, Arkadiusz Smagala, Kateryna Lyubitska

**Affiliations:** 1Department of Applied Mechanics, Mechanical Engineering Faculty, Lublin University of Technology, 20-618 Lublin, Poland; 2Polish Bearing Factory, 23-204 Krasnik, Poland; 3Department of Natural Sciences, Mid Sweden University, 851 70 Sundsvall, Sweden; 4Applied Mathematics Department, National Technical University “Kharkiv Polytechnic Institute”, 61002 Kharkiv, Ukraine

**Keywords:** bearing defect, diagnosis, recurrence plot, recurrence quantifications

## Abstract

This paper presents the problem of rolling bearing fault diagnosis based on vibration velocity signal. For this purpose, recurrence plots and quantification methods are used for nonlinear signals. First, faults in the form of a small scratch are intentionally introduced by the electron-discharge machining method in the outer and inner rings of a bearing and a rolling ball. Then, the rolling bearings are tested on the special laboratory system, and acceleration signals are measured. Detailed time-dependent recurrence methodology shows some interesting results, and several of the recurrence indicators such as determinism, entropy, laminarity, trapping time and averaged diagonal line can be utilized for fault detection.

## 1. Introduction

### 1.1. Fault Bearing Diagnosis

Bearing components are essential parts of various rotating machines. In many cases, the condition of a rotary machine depends on the state of the bearing. It is estimated that 30% of total machine failures are caused by faulty bearings [1]. The detection of bearing failure at an early stage is essential to prevent damage to other machine components. Therefore, in practice, monitoring the condition of bearing is very important.

Generally, bearing defects can be classified as localized and distributed faults. Localized defects include spalls and pits caused by fatigue wear or cracks resulting from improper manufacture (or assembly process). Distributed faults include waviness, roughness, misalignment of races, and off-size rolling elements. These faults may be due to manufacturing imperfections and operating conditions [2,3,4]. In some studies [5,6] a third category of bearing faults is proposed—an extended fault. Extended faults occur when successive rolling elements pass over the localized fault.

In practice, we can find various methods dedicated to a fault bearing diagnosis [7,8,9]. Methods can be classified as vibration measurements, acoustic measurements, temperature measurements, and wear debris analysis. The most popular and widely used industrial techniques for fault detection are vibration signals analysis and acoustic measurements. Tandon and Choudhury [10] presented a review of vibration and acoustic methods for the detection of bearing faults. Methods based on vibration measurement, acoustic emission and sound pressure were discussed. The vibration signal from a faulty bearing is acquired with the help of such techniques as the measurement of vibration response in time and frequency domain and shock pulse method (SPM). For acoustic response, the acoustic emission method (AE), sound pressure and sound intensity techniques are applied. The envelope analysis method for bearing defect detection based on the combination of the wavelet packet, the Hilbert transform, and the autocorrelation function is proposed by [11]. This method showed promising results with a high level of noise.

Many interesting studies on faulty bearing detection methods, bearing condition monitoring techniques, and bearing modeling with faults can be found in the literature. Singh et al. [12] presented a detailed review of modeling techniques for the prediction of the vibration response of a bearing with localized and extended faults. Four models were introduced: periodic impulse-train, quasi-periodic impulse-train, nonlinear multibody dynamic, and finite elements. McFadden and Smith [13] proposed a mathematical model bearing for a single point on the inner raceway under radial load. The model included constant inner ring velocity, load distribution and the effect of the internal geometry of the bearing. It was assumed that excitation would be represented as a series of repeated impulses when the rolling element hits the fault point. Such impulses can lead to bearing and machine resonances.

Liu and Gua [1] developed a deep groove ball-bearing model with a localized fault on the outer raceway. The combination of time-varying displacement excitation and time-varying contact force was presented to better understand the rolling bearing fault mechanism.

Eftekharnejed et al. [14] presented a comparison between the effectiveness of acoustic emission and vibration measurement methods for monitoring and detecting faulty rolling element bearings. The study was conducted on a test rig with a single thrust ball bearing. The data acquired from both methods were studied with the help of spectral kurtosis and a kurtogram. It was found that the acoustic emission was more sensitive to the detection of incipient faults.

Bastami and Vahid [15] investigated statistical characteristics such as RMS, peak, crest factor, kurtosis and level crossing rate in relation to the size of the bearing fault. A relation between different fault defect sizes and statistical features was established. Moreover, it was shown that the crossing rate was the most effective in monitoring rolling bearings. The results were validated with empirical data obtained from a bearing with real faults.

Wang and Laing [16] improved kurtosis by introducing a new method called “spectral kurtosis”. The proposed method determines the center frequency and bandwidth and applies the frequency domain window length for signal processing.

Junsheng et al. [17] presented an approach for detecting bearing faults based on the empirical mode decomposition method and autoregressive model for roller bearings. Empirical mode decomposition is used for the decomposition of nonstationary signals from bearings into a number of stationary mode components. For each of these components, an autoregressive model is established. A combination of the empirical mode decomposition method and the autoregressive model with the Mahalanobis distance (as a criterion function) proved to be an effective technique for detecting roller bearing failure.

Jiang et al. [18] proposed the improved variational mode decomposition based on the traditional variational mode decomposition and the empirical mode decomposition to detect the bearing fault in its early stage. This method is able to reveal the weak transient impulses from complex vibration signals.

Abbasion et al. [19] proposed a method based on wavelet analysis and Support Vector Machine (SVM) for multipoint fault detection. The authors developed an SVM network, trained it and conducted an empirical analysis of the electric motor with a combination of healthy and faulty bearings. Wavelet analysis and the proposed SVM algorithm showed satisfactory correctness for bearing fault classification. A similar method was presented by Nikolaou [20], where criteria for critical parameter selection were proposed.

Another interesting method used in the industry is the Shock Pulse Method [21]. This method gives a single value indicating the bearing condition. The disadvantage of this method is a lack of details in data interpretation. Zhen et al. [22] improved the Shock Pulse Method for more sensitive detection of faults in rolling element bearing.

In the literature, we found various methods for fault detection. The recent diagnosis methods such as Multiscale Convolutional Capsule and Self-Adaptation Graph Attention Network are described in [23,24]. A novel diagnostic approach for the possible incipient stator/rotor winding faults is proposed by [25].

Classical methods for damage detection (Fourier analyses, wavelet transformations) are nonetheless best-suited to stationary signals with high sampling rates and low noise. However, real signals are usually nonlinear and nonstationary and contain a lot of noise and interference; therefore, they are very difficult to use directly as input data for fault detection and need to be transformed by certain signal processing.

The recurrence method can also be used for bearing analysis. For example, Bo et al. [26] presented an intelligent bearing diagnosis method based on the extraction of nonlinear features from vibration signals. The combination of recurrence quantification analysis and the Kalman filter for bearing degradation evaluation is proposed by [27]. The autoregression model was built using entropy extracted from the recurrence plot.

Based on our experience, the recurrence method seems very promising for damage detection [28,29] and can produce reliable results and allow noise reduction. In this paper, the recurrence methods and particular time-dependent recurrence approach allow one to select recurrence indicators for fault indicators.

### 1.2. Motivation and Aim

A co-author of the paper is employed with a company that produces bearings. He is responsible for the detection of defective bearings. His experience and literature review show that there are no universal methods that can unambiguously diagnose damaged bearings (especially with minor faults). The classical methods applied in practice are usually useful when the signal is stationary with high sampling and low noise or when the bearing has a serious fault, which is reflected in the amplitude signal. The most sensitive methods are quite complicated and labor-intensive. In this paper, we focus on fault detection from short-measured vibration signals by simple calculation.

## 2. Materials and Methods

### 2.1. Experimental Setup

The experimental laboratory rig for measuring bearing vibration is shown in Figure 1. It consists of a main board (1), which is mounted on an anti-vibration body (2). Moreover, the system is separated from the ground by anti-vibration feet. A hydrodynamic spindle (3) is fixed to the main board. A pin (4) is clamped in the spindle.

The tested bearing (5) is placed in an adjustable positioning pocket (6) and is mounted on the pin. The positioning pocket is moved by a servo drive (7). The proper axial thrust is provided by an electric actuator (8). The axial pressure force is transferred to the tested bearing through a three-arm head (9) suspended on a ball joint. The three-arm head ensures even pressure around the entire circumference through a three-point contact with the face of the outer ring and compensates for the spindle and clamp alignment error. The spindle rotational speed is realized by the servo drive, with the controller and belt transmission located outside the main board.

A PSV3 electrodynamic vibration sensor (10) is placed in an AAF-10A measurement head (11). The sensor used in the experiment was specially designed for the laboratory rig and measures vibrations in the frequency range of 50 Hz to 10 kHz. The measuring head is located in a group of feeders positioning a head (12) in two axes. The vibration measurement with the sensor is performed by the sensor deflection unit (13) after sliding the tested bearing onto the spindle. The analog signal is amplified by the measuring amplifier and converted into a digital form by the PCIE-1802L measurement ADVANTECH card.

The measurement card was placed on an industrial PC with MS Windows 10. The software for data acquisition, control and visualization of results was written in the DELPHI environment using the Advantech DAQNavi Device Drivers libraries and the SDL Component Suite from Epina GmbH.

The method of mounting the bearing is shown schematically in Figure 2. The outer ring is stationary and loaded with a contact force. A vibration sensor is fixed above the outer ring surface. The inner ring rotates with a spindle speed.

This system allows for testing various types of bearings for various operating conditions. In our analysis, the rotational speed was maintained constant at 1800rpm.

### 2.2. Recurrence Method

Recurrence is a fundamental property of various dynamical systems, which characterize the system’s behavior. In general, recurrence is defined as all times when a dynamical system’s phase space trajectory passes over the same place in a phase space. Recurrence analysis is performed in a phase space, which is reconstructed by delayed vectors using Takens’ theorem. A measured experimental signal consisting of a sequence of scalar data $x_{1}$, $x_{2}$, *…*,$x_{n}$ can be transformed into a state vector by calculating an embedding dimension *m* and a time delay (lag) *d* according to the following equation
(1)$$X_{i}=\left(x_{i},x_{i+d},\ldots ,x_{i+(m-1)d}\right)\;i=1,2,\ldots ,n-\left(m-1\right)d.$$

The embedding dimension demonstrates the number of lagged coordinate vectors, while the time delay characterizes the length of the lag. The embedding parameters can be estimated in several ways, such as correlation dimension or neural networks. However, the most popular way to calculate them is to use the false nearest neighbors (FNN) and average mutual information (AMI) methods.

A recurrence plot (RP) is a tool used for graphical interpretation of the recurrence state, which is theoretically specified as the matrix
(2)$$RP_{i,j}=\theta (\varepsilon -||X_{i}-X_{j}||),$$
where θ stands for the Heaviside function, ∥•∥ is the norm (typically Euclidean or maximal), ε is the tolerance parameter (threshold), and $X_{i}$, $X_{j}$ are the delayed vectors of some embedding dimension. RP is a graphical representation of Equation (Equation 2), where columns and rows correspond to a certain pair of recurrence times. The final result of RP is unique, with dotted-line structures (single dots, vertical and horizontal lines) that represent the system’s dynamics and are important elements because they reveal typical dynamical features. Short line segments parallel to the main diagonal are essential features of a RP, suggesting that states are similar at different times and that the process is deterministic. If there are diagonal line structures beside single isolated points, the process is close to chaotic.

For a statistical description of the recurrence diagram structure, Recurrence Quantitative Analysis (RQA) was used. In the literature, some statistical parameters based on diagonal and vertical lines have been presented. One of the most common parameters used for diagonal line measures is determinism (DET), which characterizes the system’s predictability. The structure distribution of the vertical line is characterized by laminarity (LAM). Other recurrence quantifications are shown in Table 1.

Both recurrence methods are useful for nonlinear time series and are appropriate for analyzing minor changes in the dynamics of a complex system. The main important advantage of the recurrence method is that it can be applied to short and nonstationary data. Additionally, it does not make any assumptions about the data distribution and data size. The RP and RQA methods are based on relatively simple calculations and can be applied to a wide range of systems. Several additional indicators can be extracted to describe the characteristic of the signal.

### 2.3. Fault Modeling

Outer and inner rings, balls and a cage are the components of a rolling bearing. The rolling elements are positioned between the inner and outer rings in the cage, which keeps them in place and prevents them from colliding. Usually, the outer ring is held stationary where the inner ring and the balls rotate.

A great majority of defects are in the form of cracks, distributed defects, such as roughness or misaligned races, and deep scratches or pits that occur on the inner or outer rings during normal usage. Moreover, defects are often located in the load zone. Defects on the inner ring can occur anywhere due to rotation. These defects can cause the vibration level to increase [10]. When the rolling element is in contact with the defect, a vibration impulse is generated.

In the experiment, we used four new commercial single-row deep groove ball bearings (no. 6208C3) from the Polish Bearing Factory (PBF). The bearing has an 80 mm outer diameter, 40 mm bore diameter, and 18mm height. Bearing defects were assumed to be a local phenomenon, and thus, the real fault was stimulated (single-point defect). First, all new bearings were tested.

After that, three bearings were dismantled, and faults in the form of scratches were artificially introduced on the surface of the ball, outer ring, and inner ring. The defects were made on the bearing components by the electron-discharge machining method on the electro-erosion machine with the copper electrode. The outer and inner ring scrapes were 5 mm long and 0.5 mm deep, while the ball’s scrape was 2 mm, and the depth was 5 mm. Images of the defects (scrapes) are shown in Figure 3.

The total procedure of the bearing fault detection is shown in Figure 4. Firstly, the faults on the bearing components were introduced, and then the vibration tests were carried out.

Each measured time series was subjected to the process of normalization. Finally, RP and RQA methods were applied, and recurrence indicators for fault detection were selected.

## 3. Results and Discussion

### 3.1. Measured Time Series

Figure 5a–d shows the time velocity responses obtained in the bearing tests. The velocity vibration signals were measured in the laboratory system presented in Figure 1. In all tests, the axial force load was set to 310 N (see Figure 2), and the rotating velocity was set to 1800 rpm (reference speed). The sample rate was set to 25.6 kHz. The total measuring time for each test was 2.6 s.

It can be observed that the bearing without a fault (Figure 5a) and the bearing with a fault on the inner ring (Figure 5c) have the smallest vibrations. The amplitude in both cases is not greater than 500 μm/s. However, oscillations are substantially larger for the faults on the ball (Figure 5b) and outer ring (Figure 5d). This means that when faults occur on the ball or outer ring, the bearing’s dynamical behavior changes. It can be seen that the characteristics of the vibration impulses resulting from the defect on the inner or outer ring differ.

In panel (d), the velocity time history shows that periodic impulse signals (peaks) with a certain frequency caused a vibration pattern produced by the defect on the inner ring. The strong impulses repeat every 0.01 s (magnified in Figure 5d). However, this effect is not observed for the defects on other bearing components.

For the recurrence analysis, a normalizing process was used due to varying levels of vibration amplitudes. This procedure allows for a comparison of different signals. The measured vibration velocity signals (*v*) were normalized to zero-mean (<>) and standard deviation (σ) according to the following equation
(3)$$v=\frac{v-<v>}{\sigma }.$$

The normalization of vibration signals leads to the scaling of signals to the same vibration level, thus removing any potential disadvantages caused by the varying vibration amplitudes. The results of normalization are shown together with the recurrence diagrams in the next section.

### 3.2. Recurrence Plot Analysis

An extensive recurrence analysis is proposed to find the dynamic changes in the studied signals. The main idea is to construct RP diagrams and calculate the RQA parameters for defective and healthy bearings and, finally, compare the RQA results for the two states.

The first step in the construction of the recurrence diagram is the determination of the lag *d* and the embedding of dimension *m* parameters. Selecting the embedding parameters is a very important step in RP reconstruction and can be compared to a *focusing camera*. The AMI function is commonly used for calculating parameter *d* (Figure 6b). It is usually considered the first minimum of a mutual information function [34]. The embedding dimension *m* was prepared by FNN (Figure 6a). This method is based on measuring the percentage of near neighboring points in a given dimension. Both methods are accurately described in [28,31]. The embedding dimension *d*, the lag *m* and the threshold ϵ are given in Table 2.

For each case, the embedding dimension has the same value of *m* = 6. However, the lag *d* changes. For the bearing with defects (except for the inner ring defect case), the time lag value is reduced. For the recurrence diagram, the optimum recurrence rate is 2%; hence, the threshold parameter ϵ was calibrated to nearly 1σ (Table 2). Naturally, the threshold parameter meets the topological criterion quotes given in [35].

For various fault locations, we obtained RP diagrams that already exhibit specific features. Figure 7a–d presents the normalized time series along with the corresponding recurrence diagrams obtained for the bearing tests without defect (Figure 7a), with ball defect (Figure 7b), with inner ring defect (Figure 7c), and outer ring defect (Figure 7d).

For all cases, the recurrence diagrams were generated for 5000 data points (approximately 0.2 s), in which the first 10,000 points were rejected (data points from 10,000 to 15,000 were taken for analysis). As we can see, the recurrence topologies differ from the periodic behavior (long diagonal lines) because of multiple incommensurate frequencies. The recurrence typologies for the healthy bearing and the bearing with an inner ring defect are similar (Figure 7a,c). The RP structure of the bearing signal with a ball fault consists of darkened and white squares (Figure 7b). Small black squares with rounded corners are caused by local extremes with low amplitude vibrations. The recurrence plot in Figure 7d contains many horizontally and vertically marked spreads and white crossing lines. This means that intermittent states occur very frequently in the RP diagram and are visible in the oscillation signal. The oscillations occurring rapidly during the intermittent state disturb the bearing stabilization, causing repeated rises and drops in the amplitude of the vibration caused by the fault.

In conclusion, the recurrence diagrams exhibit different structures and provide information about the system dynamics. However, it is difficult to draw unambiguous and clear conclusions only based on the visual observation of the recurrence structures. It can be seen that the recurrence diagrams obtained for the bearing with a defect have more dark square regions. Therefore, the recurrence diagrams require a detailed statistical analysis.

An effective interpretation of RP diagrams is a recurrence quantification analysis, which reveals minor changes in the dynamics of a complex system. For this purpose, a time-dependent RQA methodology (sliding windowing technique) is useful [28]. The RP diagram is divided into overlapping small segments (windows), and RQA statistical indicators are calculated separately. Typically, this method has two variants [36]:The recurrence plot is covered by a small overlap window of size *w* that slides with steps *s*,The time signal is divided into overlapping segments, from which the RP diagrams and RQA indicators are calculated.

In this study, the first method was applied, and the analysis was made by the CRQA subroutine in the CRP toolbox [37]. The moving windows size *w* = 500 was moved along the diagonal with a step of *s* = 1. This means that 5000 data samples were used to calculate recurrence indicators. The vertical coordinate indicates the value of the computed recurrence quantification calculated over a time window. The blue line denotes the results for the bearing without fault, the red line is the results for the bearing with an inner ring fault, the green line marks the results for the bearing with an outer ring fault, and the black line represents the bearing with the ball defect.

An analysis of the results in Figure 8 demonstrates that five of twelve recurrence quantifications, i.e., DET (Figure 8a), ENT (Figure 8b), LAM (Figure 8c), TT (Figure 8d) and *L* (Figure 8g), differ in terms of their values. Their recurrence values are significantly greater, and these indicators can be used for bearing defect detection.

However, the defects in the outer ring and the ball have the smallest values of recurrence indicators. These comparable recurrence measures reveal similar results because the black areas contain diagonal and vertical lines. The number of lines depends on the defects. Other recurrence quantifications (Figure 8e,f,h–l) cannot be used for defect detection. Their values are very close in most cases. The recurrence indicators for the bearing with the ball defect exhibit the smallest values, and the recurrence diagram (Figure 7b) contains the largest amount of white regions.

Note that in our previous study on defect detection in composite machining [29], it was found that DET and LAM were the most distinctive recurrence parameters. However, a recurrence analysis was used in other dynamical processes. RP and RQA were used to determine the size and location of local defects in milling and drilling [28,29].

## 4. Conclusions

In this study, the recurrence plot and recurrence quantification methods were proposed for defect detection in bearing components.

The defect was modeled as a scratch on different bearing components, namely, the outer and inner rings and the ball. The results showed that the defect led to increases in velocity signal amplitudes and could generate characteristic periodic impulses. That was especially visible for the bearing with a defect on the outer ring. The defect in the inner ring was the hardest to diagnose, probably due to the rotation of the inner ring. Some of the recurrence quantifications achieved lower values for damaged bearing; therefore, they can be used as fault dynamic indicators. It was shown that the most promising recurrence quantifications were DET, LAM, TT, ENT and *L*. These selected recurrence quantifications showed high agreement with the results in [29].

The results are of practical significance and can be applied in real defect detection. However, the main problem is how to estimate the appropriate value of recurrence quantifications for clear defect determination. This requires further research and analysis. Therefore, in the next study, we will investigate the effect of the size of the damage on the recurrence detectability and develop appropriate algorithms.

## Figures and Tables

**Figure 1 materials-15-05940-f001:**
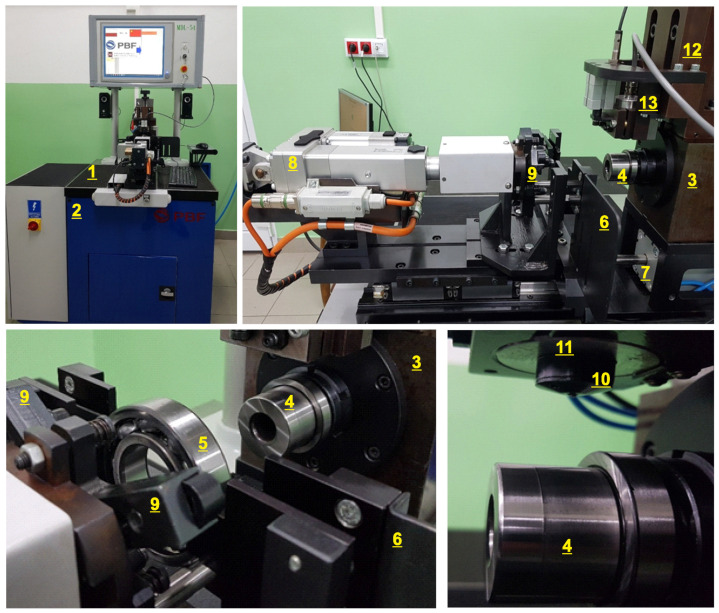
Laboratory rig for dynamic tests of bearings. The laboratory system is located at the Polish Bearings Factory in Krasnik.

**Figure 2 materials-15-05940-f002:**
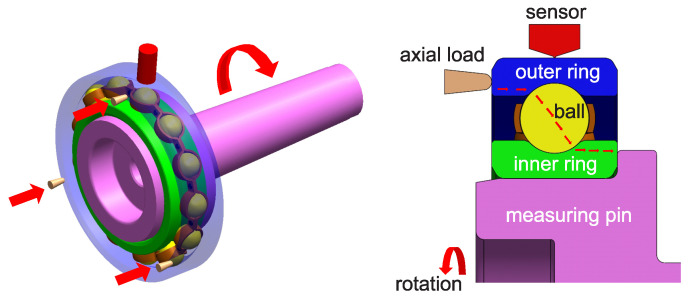
Schematic of the mounting of the bearing in the laboratory system.

**Figure 3 materials-15-05940-f003:**
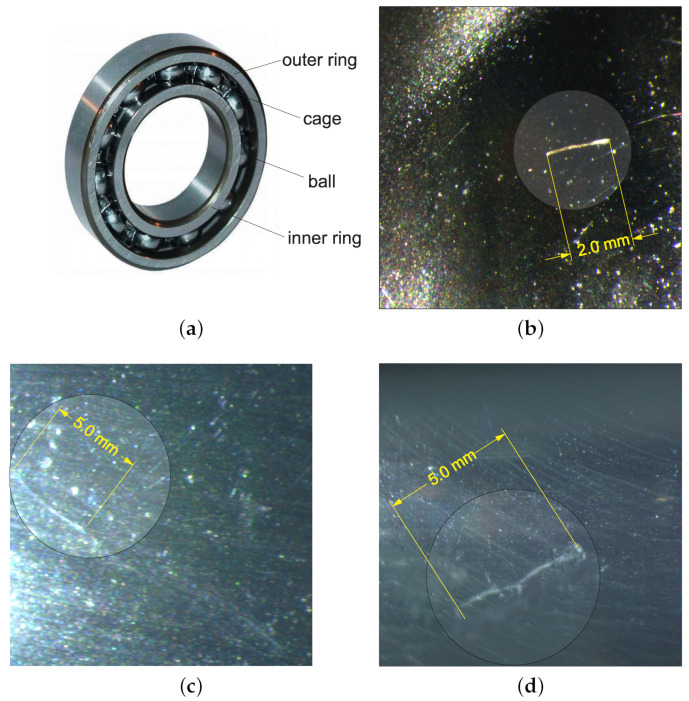
Images of rolling bearing no. 6208C3 (**a**), artificial fault on the ball (**b**), fault on the inner ring (**c**) and fault in the outer ring (**d**).

**Figure 4 materials-15-05940-f004:**
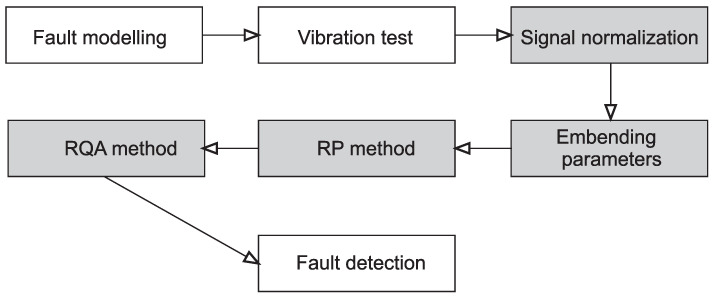
The framework of bearing fault detection. The shaded blocks show recurrence stages.

**Figure 5 materials-15-05940-f005:**
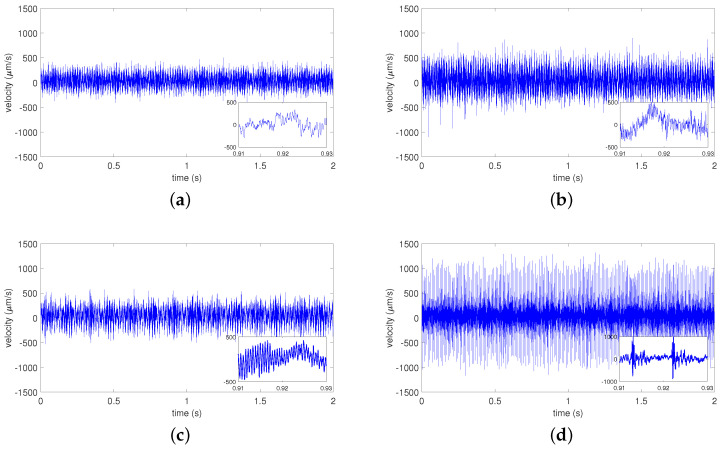
Measured time series for the tested rolling bearings: bearing without fault (**a**), bearing with ball fault (**b**), bearing with inner ring fault (**c**), and bearing with outer ring fault (**d**).

**Figure 6 materials-15-05940-f006:**
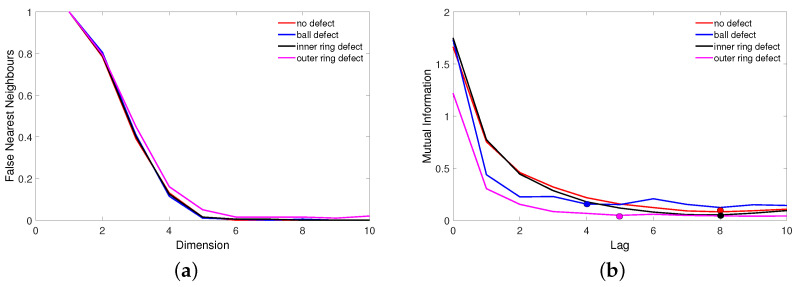
Results of FNN (**a**) and AMI (**b**) methods for estimating embedding parameters *m* and *d*. The points represent the estimated optimal lag values.

**Figure 7 materials-15-05940-f007:**
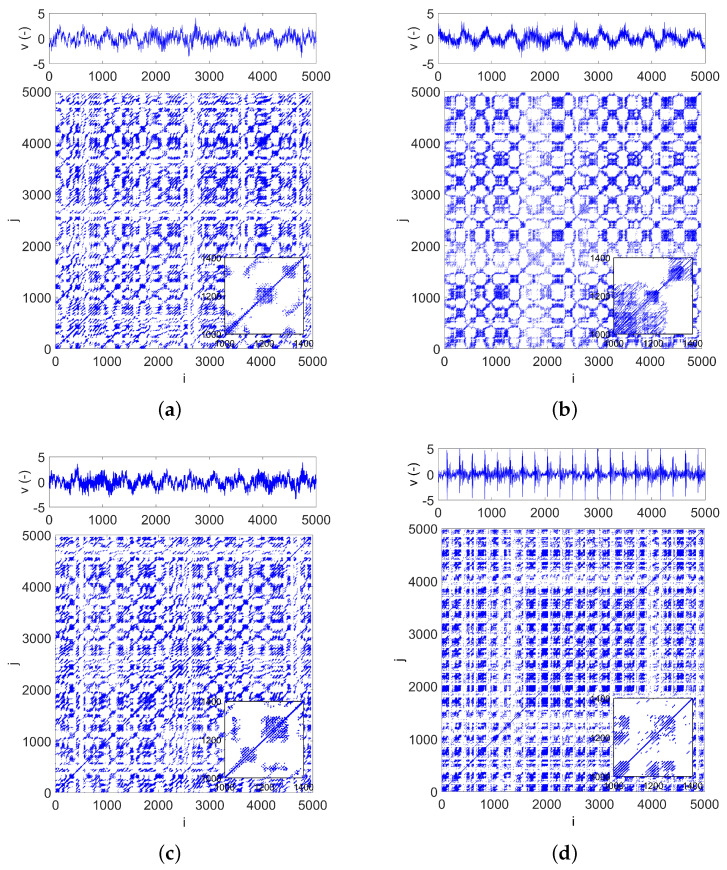
Recurrence plots calculated for the bearing: without defect (**a**), with ball defect (**b**), with inner ring defect (**c**) and outer ring defect (**d**).

**Figure 8 materials-15-05940-f008:**
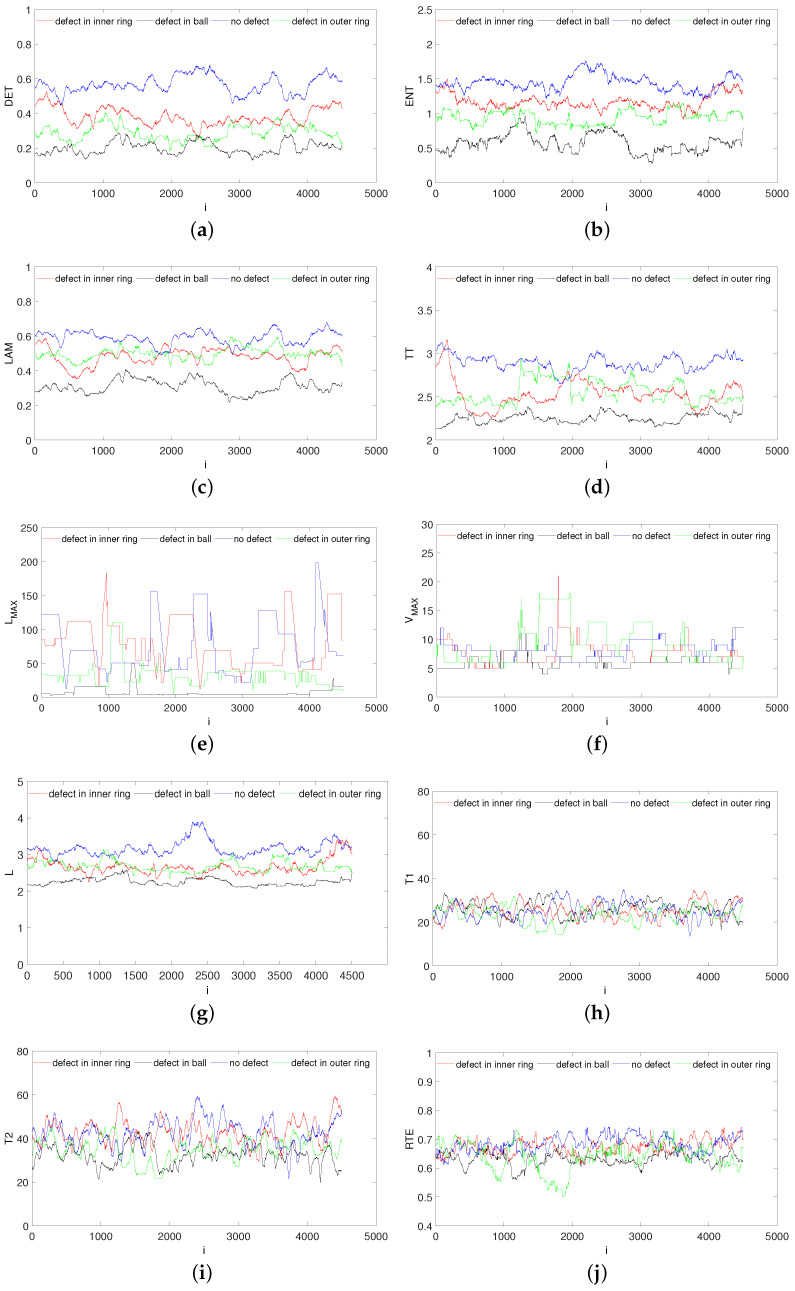
Recurrence quantifications versus shifting time window: DET(**a**), ENT (**b**), LAM (**c**), TT (**d**), $L_{MAX}$ (**e**), $V_{MAX}$ (**f**), L (**g**), T1 (**h**), T2 (**i**), RTE (**j**), Trans (**k**) and Clust (**l**).

**Table 1 materials-15-05940-t001:** Definition of the most useful recurrence quantifications [28,30,31,32,33]. P(l) and P(v) denote the distribution of the length of diagonal and vertical lines. $N_{l}$ and $N_{v}$ are the numbers of diagonal and vertical lines. $R_{i}$ is the recurrence point that belongs to the state $\bar{x_{i}}$, and Hv(v) is the distribution.

Quantification	Equation	Description
Recurrence Rate RR	$\frac{1}{N^{2}}$ $\sum_{i,j=1}^{N}RP_{i,j}$	Recurrence point density.
Determinism DET	$\frac{\sum_{l=l_{min}}^{N}lP\left(l\right)}{\sum_{i,j=1}^{N}RP_{i,j}}$	Portion of recurrence points forming diagonal lines.
Entropy ENT	$-\sum_{l=l_{min}}^{N}P\left(l\right)ln\left(P\left(l\right)\right)$	Entropy of the frequency distribution of the diagonal lines.
Laminarity LAM	$\frac{\sum_{v=v_{min}}^{N}vP\left(v\right)}{\sum_{v=1}^{N}vP\left(v\right)}$	Amount of recurrence points that form vertical lines.
Trapping Time TT	$\frac{\sum_{v=v_{min}}^{N}vP\left(v\right)}{\sum_{v=v_{min}}^{N}P\left(v\right)}$	Average length of vertical lines.
Longest diagonal line $L_{max}$	$max(\left\{l_{i};i=1,...,N_{l}\right\}$	Maximal line length in the diagonal direction.
Longest vertical line $V_{max}$	$max(\left\{v_{i};i=1,...,N_{v}\right\}$	Maximal length of the vertical structures.
Averaged diagonal line *L*	$\frac{\sum_{l=l_{min}}^{N}lP\left(l\right)}{\sum_{l=l_{min}}^{N}lP\left(l\right)}$	Average diagonal line length.
Recurrences time T1	$|\left\{i,j:\;\bar{x_{i}},\bar{x_{j}}\right\}\in R_{i}\}|$	Recurrence time of the 1st Poincare recurrence.
Recurrences time T2	$|\left\{i,j:\;\bar{x_{i}},\bar{x_{j}}\right\}\in R_{i},\bar{x_{j}}\notin R_{i}\}|$	Recurrence time of the 2nd Poincare recurrence.
Recurrence time entropy RTE	$-\frac{1}{lnV_{max}}\sum_{v=1}^{V_{max}}H_{v}\left(v\right)lnH_{v}\left(v\right)$	Shannon entropy of the recurrence times.
Transitivity Trans	$\frac{\sum_{i,j,k=1}^{N}RP_{i,j}RP_{j,k}RP_{k,i}}{\sum_{i,j,k=1}^{N}RP_{i,j}RP_{k,i}}$	Local recurrence rate.
Clustering coefficient Clust	$\sum_{i=1}^{N}\frac{\sum_{j,k=1}^{N}RP_{i,j}RP_{j,k}RP_{k,i}}{RR_{i}}$	The probability that two recurrence states are neighbors.

**Table 2 materials-15-05940-t002:** Embedding parameters estimated by FNN and AMI methods.

Location of Defect	Embedding Dimension, m	Lag, d	Recurrence Rate, RR	Threshold, ϵ
No defect	6	8	0.02	0.88
Ball	6	4	0.02	0.80
Outer ring	6	5	0.02	0.61
Inner ring	6	8	0.02	0.88

## Data Availability

Not applicable.

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
