# Peer review of "Ball Bearing Fault Diagnosis Using Recurrence Analysis"

_materials, 2022, doi:10.3390/ma15175940_

Round 1

Reviewer 1 Report

Improve the abstract by adding the findings

For fault modelling, how did the faults in the form of scratches were artificially produced on the surface of the ball, outer ring, and inner ring?

Why the DET and LAM are concluded as the most distinctive recurrence parameters compared to the ENT and TT in bearing fault diagnosis? How to access quantitatively?  

Reviewer 2 Report

The novelty of the approach is significant and the manuscript has archival value. The results are acceptable given the difficult task to carry out.

This reviewer misses some technical data about the sensing device, the accelerometer.

Reviewer 3 Report

.

Reviewer 4 Report

This paper focuses on the rolling bearing fault diagnosis using recurrence analysis. It is of interest. However, the authors should address the following points to further improve the quality.

1. Motivation and contribution should be summarized in the abstract. Currently, the abstract is too simple and general.

2. The contributions of this paper should be highlighted at the end of the introduction. Why the classical methods cannot be applied for faulty bearings whose signals are very similar to the signals from normal bearings?

3. What’s the difference between the recurrence method and the other signal processing methods?

4. It is better to give a detailed procedure to describe the whole framework of the proposed fault diagnosis approach.

5. Literature review on the approaches of fault diagnosis is limited. More recently-published papers in this field should be discussed. The authors may be benefited by reviewing more papers such as 10.1016/j.ymssp.2022.109569 and 10.1109/TIM.2022.3181894.

6. There are many deep learning networks for fault diagnosis. What’s the superiority of the proposed approach?

Reviewer 5 Report

The problem of rolling bearing fault diagnosis based on vibration velocity signal is presented in this paper. Overall, the paper is well written and organized with a proper length. The contributions as well as the quality are both good. In addition, there are some points that are not very clear and should be addressed in the revised version:

1. Please update the references, especially for recent years. For example, incipient fault diagosis is an important research issue in the field of rotating machinery. The authors should supplement some results on this aspect, for example the following references had given significant design results:

[1] Incipient winding fault detection and diagnosis for squirrel-cage induction motors equipped on
CRH trains
. ISA Transactions, 2020, 99: 488~495.

2.  The innovation of this paper is not clear and it is difficult for readers to understand the main contributions of this paper. This part should be added in Introduction section.

3.   The description of the existing work should be shorter in Introduction section.  Furthermore, more descriptions of the proposed method are needed.

4.   The difficulties in signal processing-based fault diagnosis are the system noise (The mechanical vibration will have influnces on diagnosis accuracy). How to solve this problem? Please give a remark. 

Round 2

Reviewer 4 Report

The revision has addressed all my issues. The quality has improved a lot after revision. I recommend it for publication.

Reviewer 5 Report

In a general way most of my comments were answered by the authors. My overall opinion about this paper is quite good. The manuscript is well written and acceptable for publishing